# Serial Anti-GM-CSF Autoantibody Levels Reflect Disease Activity in Hypersensitivity Pneumonitis with Autoimmune Pulmonary Alveolar Proteinosis: Case Report

**DOI:** 10.3390/pathophysiology32030047

**Published:** 2025-09-15

**Authors:** Toru Arai, Masaki Hirose, Eiji Sugimoto, Takayuki Takimoto, Yoshikazu Inoue, Hiromitsu Sumikawa, Tamiko Takemura, Shigeki Shimizu

**Affiliations:** 1Clinical Research Center, NHO Kinki Chuo Chest Medical Center, Sakai City 591-8555, Osaka, Japan; hirose.masaki.tb@mail.hosp.go.jp (M.H.); takimoto.takayuki.ra@mail.hosp.go.jp (T.T.); giichiyi@me.com (Y.I.); 2Department of Cancer, Infection and Immunology, Graduate School of Medicine/Faculty of Medicine, Osaka University, Suita City 565-0871, Osaka, Japan; 3Department of Respiratory Medicine, NHO Kinki Chuo Chest Medical Center, Sakai City 591-8555, Osaka, Japan; dcp535cn@yahoo.co.jp; 4Department of Internal Medicine, Osaka Anti-Tuberculosis Association Osaka Fukujuji Hospital, Neyagawa City 572-0850, Osaka, Japan; 5Department of Radiology, NHO Kinki Chuo Chest Medical Center, Sakai City 591-8555, Osaka, Japan; h-sumikawa@radiol.med.osaka-u.ac.jp; 6Department of Pathology, Kanagawa Cardiovascular and Respiratory Center, Yokohama City 236-0051, Kanagawa, Japan; tamikobyori@gmail.com; 7Department of Pathology, NHO Kinki Chuo Chest Medical Center, Sakai City 591-8555, Osaka, Japan; shimizush@hotmail.com

**Keywords:** autoimmune pulmonary alveolar proteinosis, granulocyte-macrophage colony-stimulating factor, hypersensitivity pneumonitis

## Abstract

Autoimmune pulmonary alveolar proteinosis (aPAP) is characterized by the accumulation of phospholipids and surfactant proteins in the peripheral air spaces due to alveolar macrophage dysfunction caused by anti-granulocyte-macrophage colony-stimulating factor (GM-CSF) autoantibodies (GMAb). Hypersensitivity pneumonitis (HP) is a granulomatous lung disease associated with GM-CSF. In this report, we evaluated serial changes in serum GMAb levels in a 67-year-old male current smoker with HP and aPAP and examined their correlation with HP disease activity. GMAb levels increased at HP onset and decreased after HP remission with oral prednisolone therapy. After the first remission, the patient experienced three relapses and remissions. Although GMAb levels were not evaluated for all HP relapses and remissions, GMAb levels increased at one relapse but decreased at two remissions induced by the oral prednisolone therapy. Pulmonary fibrosis progressed, and the patient died of pneumonia. GMAb was at its almost normal levels at 8 months before the onset of pneumonia. We hypothesized that GMAbs may have been induced to improve HP through neutralizing GM-CSF. Although the hypothesis needs to be confirmed in additional patients, serial measurement of GMAb may be useful for a better understanding of the pathophysiology and deciding the appropriate treatment for HP with aPAP.

## 1. Introduction

Autoimmune pulmonary alveolar proteinosis (PAP) (aPAP) is characterized by the accumulation of phospholipids and surfactant proteins in the peripheral air spaces due to alveolar macrophage dysfunction caused by anti-granulocyte-macrophage colony-stimulating factor (GM-CSF) autoantibodies (GMAb) [1,2]. Hypersensitivity pneumonitis (HP) [3] is a granulomatous lung disease associated with GM-CSF [4,5], and several patients with HP complicated by aPAP have been reported [6,7]. Although Verma et al. reported five patients with both HP and PAP, GMAb were negative in two patients and not measured in the remaining three patients [8]. Here, we report the first case of HP and aPAP in which serum GMAb levels were serially evaluated using an enzyme-linked immunosorbent assay [9] and examined the correlation with HP activity. All GMAb levels were measured by the same method.

## 2. Case Report

A 67-year-old male, a current smoker, presented to our hospital 6 months after the first detection of abnormal chest shadows during a health check. He worked as an electrician for 10 years in his 20s, and it is undeniable that he was exposed to asbestos. The shadows had not been detected by the previous health check. He did not use a feather duvet, and he had no experience raising birds. He had no cough and no shortness of breath (SOB) on his first visit to our hospital. Chest computed tomography (CT) showed a reticular nodular opacity, patchy ground-glass opacity (GGO) (Figure 1 and Figure 2A), and three density patterns, which were compatible with the HP pattern [3]. Pleural plaque suggesting asbestos exposure was not noted. Cell analysis of the bronchoalveolar lavage (BAL) fluid showed >30% lymphocytosis. BAL cytology findings indicated PAP (Figure 2), and periodic acid-Schiff (PAS) stain-positive proteinaceous material was observed (Figure 2D). Blood tests showed elevated serum GMAbs (51.2 μg/mL; cutoff value, 3.33 μg/mL [9]) and Krebs von den Lungen-6 (KL-6) [10] (6480 U/mL; cutoff value, 500 U/mL) (“A” in Figure 3). Serum autoantibodies suggesting connective tissue diseases were not detected. Fibrotic HP (FHP) complicated by aPAP was suspected, and surgical lung biopsy (SLB) was performed to confirm the diagnosis. Histological examination of SLB specimens (Figure 4A,B) revealed lymphocyte infiltration in the alveolar walls, alveolar ducts, and respiratory bronchioles, in addition to predominant subpleural and paraseptal fibrosis. Airway-centered fibrosis was observed, and granulomas were not detected. These histological findings suggested a probable HP pattern according to the ATS/ERS/JRS guideline [3]. Dense proteinaceous material was observed in the air spaces in some areas (Figure 4C,D), and the material was PAS staining positive (Figure 5). From these findings, this patient was diagnosed with FHP with moderate confidence according to the ATS/ERS/JRS guideline [3] and was found to be complicated with aPAP.

SOB and CT findings worsened following hospital discharge after SLB, leading to acute exacerbation (Figure 6B); KL-6 and GMAb levels increased to 13,770 U/mL and 120.9 μg/mL (“B” in Figure 3), respectively. However, the SOB spontaneously improved after the patient returned to our hospital. Inciting antigens could not be specified; however, home antigens were suspected to be related to the patient’s HP activity. At this point, the patient could be diagnosed with FHP with high confidence according to the ATS/ERS/JRS guideline [3]. PAS-positive material was not observed in the second BAL during the acute exacerbation, suggesting that FHP, rather than aPAP, predominantly deteriorated. Therefore, prednisolone (35 mg/day) was administered. One month later, KL-6 and GMAb levels began to decrease simultaneously (“C” in Figure 3) and finally, before Relapse 1, decreased to 2985 U/mL and 3.16 μg/mL (lower than the cutoff level) (“D” in Figure 3), respectively. The predictive value of the forced vital capacity (FVC) and diffusing capacity of carbon monoxide increased from 66.8% and 38.9% (“B” in Figure 3) to 85.2% and 62% (“C” in Figure 3, 1 month after the initiation of prednisolone), respectively. SOB improved after the prednisolone treatment, while nodular lesions and GGO decreased (Figure 6C). After HP remission, the prednisolone dose was tapered off.

The patient experienced three FHP relapses after the first remission (Figure 3), and steroid therapy improved SOB in every FHP relapse. The patient was managed without long-term oxygen therapy until pulmonary fibrosis progressed approximately 5 years after the diagnosis. High-resolution CT at this time showed reticulation and traction bronchiectasis without honeycombing (Figure 6D). We could not evaluate the GMAb levels during the first and third relapse; however, after remission of the first and third relapse, the GMAb levels were very low: 0.60 (“E” in Figure 3) and 9.83 μg/mL (“H” in Figure 3), respectively. The levels increased to 41.3 μg/mL at the beginning of the second relapse (“F” in Figure 3) and decreased to 28.6 μg/mL after corticosteroid initiation (“G” in Figure 3). The patient developed chronic respiratory failure due to pulmonary fibrosis progression and died of pneumonia approximately 6 years after the initial hospital visit (**†** in Figure 3). We could not administer nintedanib because the patient died before the drug had proven effectiveness for progressive pulmonary fibrosis.

## 3. Discussion

Serial measurement of serum GMAb was rarely performed in the previous reports; GMAb levels in our patient increased at the onset of HP with aPAP and decreased after remission with oral prednisolone. Steroid administration to aPAP patients is known to deteriorate symptomatic, radiologic, and physiological activity of aPAP and increase serum biomarkers [11]. Hence, disease improvement after the initiation of oral prednisolone was supposed to reflect the disease activity of HP, not that of aPAP. Although GMAb levels could not be measured at every HP relapse, we observed an increase during the second relapse and very low levels during the remission after the first and third relapse following corticosteroid therapy.

Asami-Noyama et al. reported a case of HP preceding aPAP [7], in which the patient was diagnosed with FHP based on GGO and traction bronchiectasis, predominantly in the lower lobe on chest CT, and >50% lymphocytosis without PAS-positive material in the BAL fluid. After initiating immunosuppressive therapy for FHP due to FVC deterioration, PAS-positive material was observed in the transbronchial lung biopsy specimens, and the BAL fluid became cloudy. Serum GMAb was proven to be positive, and the patient was diagnosed with aPAP. Whether GMAb was present at HP onset or induced afterward remained unclear. However, unlike previous reports [7,12] where GMAb was detected after the initiation of corticosteroids, GMAb production in our patient was suppressed following immunosuppressive therapy, and aPAP did not deteriorate. Akasaka et al. also reported that serum GMAb levels in steroid-treated aPAP were similar to those in non-treated aPAP [11]. The remarkable reduction in serum GMAb in our patient after steroid therapy initiation was supposedly inconsistent with aPAP. Hence, the physiological role of GMAb in this patient might be different from that of typical aPAP.

GMAbs delicately regulate GM-CSF function in healthy individuals [13]. We hypothesized that GMAbs were induced in response to etiological antigen stimulation, aiming to neutralize GM-CSF and modulate HP activity in our patient. GM-CSF has been reported to be associated with granuloma formation in vitro and in vivo [14,15]. GM-CSF concentration in BAL fluids of HP was significantly higher than that in healthy individuals [16]. Hence, immunosuppressive treatment with corticosteroids may reduce the local levels of GM-CSF associated with HP, resulting in decreased serum GMAb levels. Based on this hypothesis, we could explain some important phenomena in addition to the reduction in serum GMAb after steroid therapy observed in our patient. Verma et al. showed that their two patients with HP and PAP showed negative serum GMAb and their PAP was secondary [8]; however, it is not clarified when GMAb was measured in these patients. We believe it remains uncertain whether serum GMAb levels of the two patients were truly always negative. GMAb could have been positive at some points during their clinical course, similarly to our patient, if serial measurement of GMAb had been performed in the two patients. Our patient showed negative GMAb during the remission of FHP after the first episode and Relapse 1 (“D” and “E” in Figure 3). To exclude the possibility of comorbid aPAP completely, it would be preferable to measure serum GMAb levels at multiple points reflecting different pathophysiological backgrounds—at diagnosis, during immunosuppressive therapy, after remission, and at later follow-up points.

In our patient, PAP was pathologically observed in a limited number of areas, particularly in the air spaces of the fibrotic lesions. The PAP findings possibly appeared in places where GMAb was locally dominant over GM-CSF, although such findings could also be observed in idiopathic pulmonary fibrosis [17]. In addition, a milky appearance and proteinaceous material aggregation, suggesting PAP, were observed in the first but not the second BAL fluid when HP was aggravated rapidly before prednisolone initiation. This may be because GM-CSF was rapidly induced by inhalation of the etiological antigen, and macrophages activated by the antigen phagocytosed the locally deposited proteinaceous materials.

GMAb production in our patient was reduced by immunosuppressive therapy, and the PAP did not worsen. If aPAP was defined as PAP resulting from uncontrolled or unregulated GMAb production, then our patient did not meet the criteria for aPAP. In contrast, the case of aPAP following HP reported by Asami-Noyama et al. likely represents “true” aPAP [7]. However, the clinical course and GMAb changes in our patient could not be predicted at the time of diagnosis, and our patient should be diagnosed with HP complicated by aPAP. As immunosuppressive treatment can aggravate aPAP [11], we carefully monitored and treated the patient for HP with coexisting aPAP. When radiological GGO and KL-6 levels increased, we used BAL fluid analysis to distinguish between HP and aPAP exacerbation and started corticosteroid therapy. If BAL findings suggested deterioration of aPAP, whole lung lavage would have been the preferred therapy [18], though GM-CSF inhalation is now favored [19]. If deterioration of FHP was suggested, prednisolone therapy should be initiated with a comparatively low dose (e.g., less than 0.5 mg/kg/day) [20]; however, aggravation of aPAP should be ruled out if the disease condition worsens after the initiation of prednisolone.

The coexistence of a granulomatous lung disease such as HP and aPAP caused by GMAb is perplexing, as GM-CSF is a key cytokine in granuloma formation [14,15]. We hypothesized that GMAb may suppress granuloma formation, offering a potential explanation for the pathophysiology in some patients. Silva et al. previously reported a case of aPAP preceding HP [6]; their patient was diagnosed with aPAP based on typical findings of chest CT and BAL fluid and positive serum GMAb. Five years after the aPAP diagnosis, Chest CT showed reticular opacity in peripheral distribution and traction bronchiectasis in the basal lungs. PAS-positive material was not observed in the BAL fluid. Histological findings of cryobiopsy specimens showed intra-alveolar eosinophilic materials suggesting PAP and coexisting findings of HP. Hence, the patient was diagnosed with FHP and aPAP after the multidisciplinary discussion. The pathophysiology of HP development after the control of aPAP has not been clarified. Silva et al. suggested that alveolar damage following PAP might have induced susceptibility to environmental triggers, leading to the development of HP. However, we suppose the hypothesis of Silva et al. [6] could not explain why FHP developed after aPAP was under control, although aPAP might create a condition susceptible to HP. We propose instead that the onset of HP might have been suppressed by GMAb when aPAP was prominent, and HP might have appeared after aPAP remission with possibly reduced serum GMAb. However, GMAb levels were not serially examined in their patient.

GMAb might be associated with the pathophysiology of other lung diseases. Sarcoidosis is another granulomatous disease pathophysiologically associated with GM-CSF [21,22]. We have reported a case of a patient with sarcoidosis complicated with aPAP. Sarcoidosis in the eyes, lungs, and skin occurred after the remission of aPAP following GM-CSF inhalation therapy [23]. At the onset of sarcoidosis, serum levels of GMAb were reduced to almost the cutoff level (3.53 μg/mL; cutoff value, 3.33 μg/mL), and this might have caused the occurrence of sarcoidosis. Etiological antigens for sarcoidosis have not been determined; however, mycobacteria and *Propionibacterium acnes* are thought to be associated [21]. We have hypothesized that macrophage dysfunction in aPAP preceding sarcoidosis caused increased accumulation and extended distribution of causative antigens. As a result, remission of aPAP and decreased GMAb levels induced the onset of sarcoidosis and multiorgan diseases. This hypothesis might also explain the mechanism by which FHP develops after the aPAP resolution. Increased deposition of allergens in the lung might make patients with aPAP susceptible to FHP, as suggested by Silva et al. [6]. We have also reported on a patient with tuberculous mediastinal lymphadenitis and aPAP [23]. After the remission of aPAP by GM-CSF inhalation therapy, tuberculous mediastinal lymphadenitis occurred, and serum levels of GMAb were approximately 10 μg/mL [23], which was similar to the threshold of GMAb levels suppressing the functions of myeloid cells, including macrophages, as reported by Uchida et al. [13]. Due to the reduction in GMAb, the granulomatous reaction to *Mycobacterium tuberculosis* was possibly activated, and lymphadenitis occurred. Additionally, GMAb could be a biomarker for diseases other than pulmonary diseases; it has been reported to predict disease severity in inflammatory bowel diseases [24].

We have used serum KL-6 levels to show the disease activity of this patient. KL-6 is a high-molecular-weight glycoprotein, classified as a human MUC1 mucin. It was first discovered as a serum tumor biomarker for pulmonary, breast, and pancreatic cancers [25]; however, further investigation has shown that it is also useful for monitoring the activity of interstitial lung diseases (ILDs) [25]. Okamoto et al. reported that KL-6 reflected the seasonal changes in disease activity in bird-related FHP [26]. Serum KL-6 levels in FHP were significantly higher than those in other ILDs [27]. It is also useful in monitoring the severity of disease and the clinical course of aPAP [28,29]. We have shown that KL-6 is highly correlated with SOB, %DLco, and %VC [10], and also reported that serum KL-6 levels in aPAP were significantly higher than those in FHP and other ILDs [10]. Hence, serum KL-6 levels >10,000 U/mL in our patient at diagnosis were tremendously high and consistent with comorbid aPAP; however, peak serum KL-6 levels decreased at disease relapse, and this might suggest that the relapse was mainly FHP, although it was not histologically examined.

This report has some limitations. First, the most important limitation was that serum levels of GMAb were not measured at the limited points, and the elevation and decline of GMAb at all the relapses and remissions of HP were not clarified. In this study, we measured serum levels of GMAb retrospectively, except at diagnosis; unfortunately, serum samples were missed at important time points. Second, local GMAb distribution in the lung could not be evaluated. However, we hypothesized the presence of an imbalance between GMAb and GM-CSF in the lung, causing local PAP findings. Examination of the local relationship between GM-CSF and GMAb by BAL fluid could not be performed. Third, we suppose that not aPAP but FHP relapsed at the three relapse events of this patient according to the efficacy of prednisolone; however, which disease relapsed was not clarified by BAL and/or lung biopsy. Fourth, a patient similar to ours has not been reported; however, this might be because GMAb could not be easily evaluated in every hospital. In addition, even if the antibody is measured, whether it will be positive or not might depend on the clinical phase of each patient, as we have reported.

## 4. Conclusions

We present a case of a patient with HP and aPAP in whom serum GMAb levels reflected disease activity. While serum GMAb levels are generally not associated with disease severity of aPAP, GMAb could affect the onset and clinical course of various diseases other than aPAP. Serial measurement of GMAb may be useful for a better understanding of the pathophysiology and deciding the appropriate treatment for HP with aPAP. Further accumulation of evidence in more patients with HP and aPAP is warranted to support our hypothesis.

## Figures and Tables

**Figure 1 pathophysiology-32-00047-f001:**
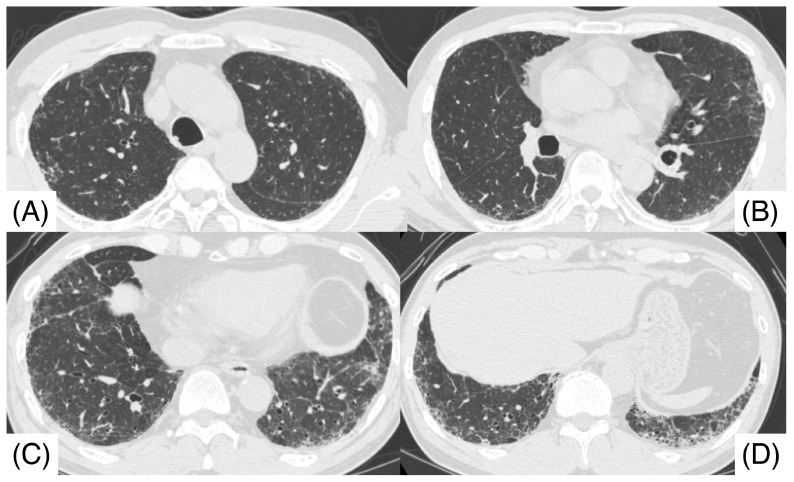
Chest computed tomography before surgical lung biopsy shows lower lobe predominant shadows (**C**,**D**), including patchy ground glass opacity, reticulonodular opacity, and nodular lesions. Centrilobular nodules were observed in both upper lobes (**A**,**B**). The computed tomography pattern of this patient was compatible with a hypersensitivity pneumonitis pattern.

**Figure 2 pathophysiology-32-00047-f002:**
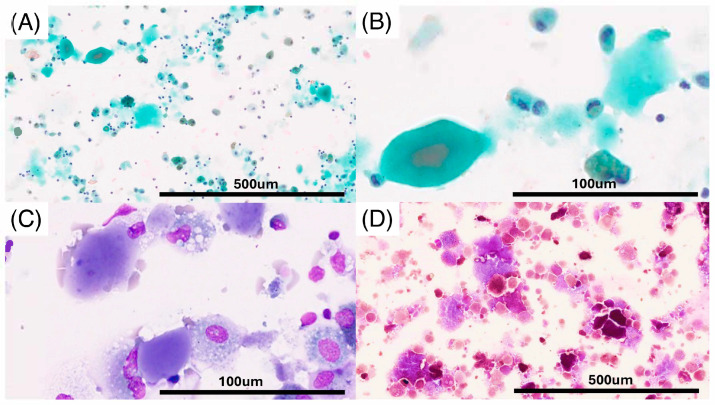
Foamy macrophages, and large and small eosinophilic granules were scattered (Papanicolaou stain; (**A**), ×100; (**B**), ×400). Giemsa stain ((**C**), ×400) and Periodic Acid Shiff stain ((**D**), ×100) was shown.

**Figure 3 pathophysiology-32-00047-f003:**
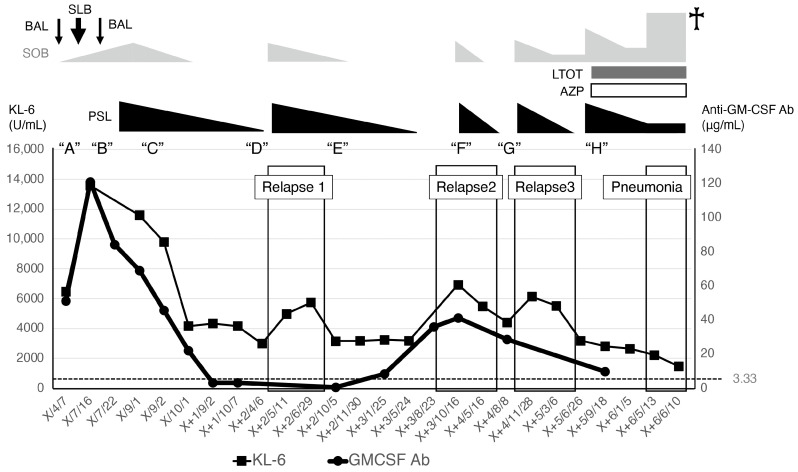
Clinical course of SOB, therapy, and serum KL-6 and anti-GM-CSF antibody (GMAb) levels are shown. The patient experienced three relapses of fibrotic hypersensitivity pneumonitis and died approximately 6 years after diagnosis (†). At the relapse 2, GMAb levels were shown to be elevated, although they were not evaluated at Relapses 1 and 3. “A”: 1st visit, “B”: acute exacerbation, “C”: one month after initiation of prednisolone, “D”: before Relapse 1, “E”: remission after Relapse 1, “F”: Relapse 2, “G”: after prednisolone initiation, “H”: remission after Relapse 3.

**Figure 4 pathophysiology-32-00047-f004:**
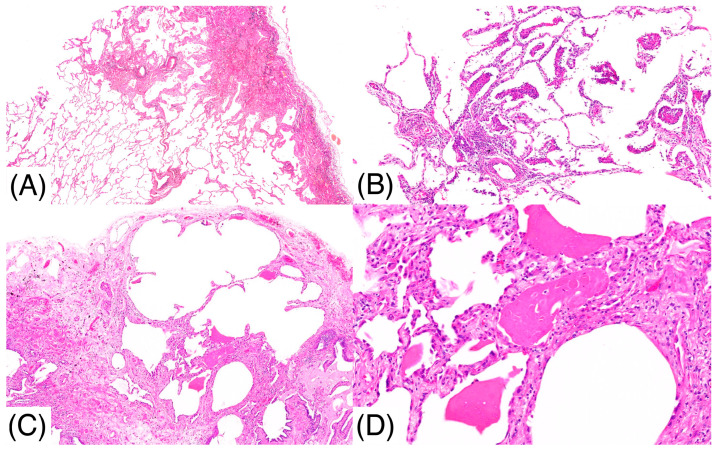
Histological examination (Hematoxylin and Eosin stain) of SLB specimens (**A**,**B**) reveals lymphocyte infiltration in the alveolar walls, alveolar ducts, and respiratory bronchioles in addition to predominant subpleural and paraseptal fibrosis with airway-centered fibrosis. Dense proteinaceous materials, suggesting pulmonary alveolar proteinosis, are also observed in the air spaces in some fibrotic areas (**C**,**D**). Panel (**D**) shows an enlarged view of the rectangular area of panel (**C**).

**Figure 5 pathophysiology-32-00047-f005:**
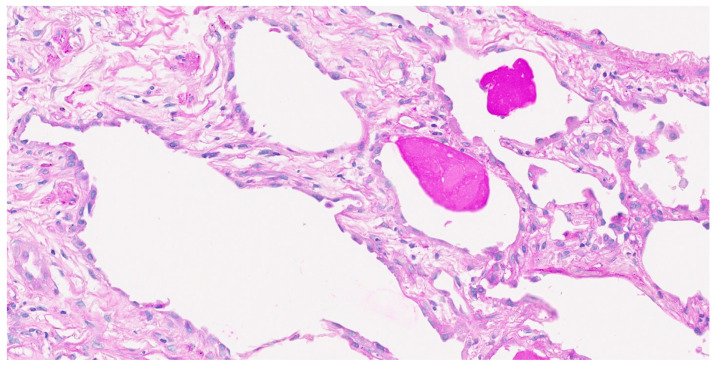
Histological examination of surgical lung biopsy specimens; periodic acid-Schiff stain-positive proteinaceous material, indicative of pulmonary alveolar proteinosis, was shown.

**Figure 6 pathophysiology-32-00047-f006:**
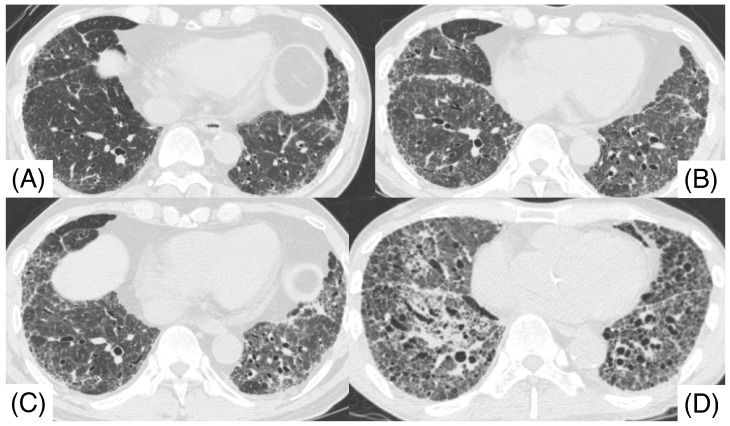
Chest computed tomography before surgical lung biopsy shows patchy ground glass opacity (GGO) and reticulonodular opacity (**A**), and the GGO is remarkably increased at the acute exacerbation before prednisolone administration (**B**). Approximately 2 months after initiation of prednisolone, GGO and nodular lesions decrease (**C**). Five years after diagnosis, traction bronchiectasis and reticular opacity progress (**D**).

## Data Availability

Data are available upon reasonable request after the approval of our institutional review board.

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
