# Peer review of "Serial Anti-GM-CSF Autoantibody Levels Reflect Disease Activity in Hypersensitivity Pneumonitis with Autoimmune Pulmonary Alveolar Proteinosis: Case Report"

_pathophysiology, 2025, doi:10.3390/pathophysiology32030047_

Round 1
Reviewer 1 Report
Comments and Suggestions for Authors
Toru Arai and colleagues presents the case of a patient for whom the diagnoses of hypersensitivity pneumonitis and autoimmune pulmonary alveolar proteinosis are assumed to coexist
The diagnosis of APAP is unambiguous: PAS-positive proteinaceous material + anti GM-CSF autoantibodies above a pathogenic threshold
It should be noted that the diagnosis of HP is less clear-cut, as the diagnostic criteria for a definite HP according to the ATS/ERS consensus were not found.
Do the authors have any details on the patient's respiratory exposures (apart from smoking), suspicion of an aeroallergenic cause?
What is the duration of the patient's symptoms and radiological abnormalities before being referred to the hospital?
Cases of coexisting HP and PAP are not numerous, and the temporality between these two entities is obviously an essential point of discussion.
Could the authors please specify which biological test they used to quantitatively determine the presence of autoantibodies. Is it the same technique at the different time points?
Was the screening for autoimmunity associated with interstitial lung disease otherwise negative?
Was there a preferential distribution of ground-glass and fibrous areas on the thoracic imaging (predominance at the lower lobes?)
Figure 3: the x-axis date legend is not clear for me
The originality of this case lies in the serial measurement of anti GM-CSF antibody levels in parallel with the patient's clinical course.
The authors' hypothesis is that elevation of anti-GM-CSF antibodies is a meant to neutralize GM-CSF over-excretion, which is thought to be involved in the pathophysiology of HP.
They assume that the decrease in antibodies cannot be explained by immunosuppressive therapy and is therefore a marker of resolution of the HP relapse.
They lack solid arguments to support their hypothesis, due to the retrospective nature of this description. Would it be possible to also measure level of free GM-CSF in BAL?
They could also discuss a direct causal link between anti-GM-CSF antibodies and HP relapses.
However, this is an attractive hypothesis, and the presentation of this case raises the question of the role of GM-CSF and anti-GM-CSF antibodies in the pathophysiology of certain types of interstitial lung disease
The role of anti-GM-CSF antibodies as a regulator of pulmonary GM-CSF levels could indeed be a means of mitigating the onset or aggravation of certain inflammatory ILD
This could open up prospects for systematically measuring GM-CSF levels in BAL and anti-GM antibody levels (blood and BAL) in the diagnosis and follow-up of HP. I am less confident about the immediate prospects in terms of treatment (“Deciding on the appropriate treatment for HP with APAP.”)
Author Response
Response to Reviewer 1
Thank you for your important comments. I have modified our manuscript according to your comments and I would like to explain how we have modified point by point.
In addition, a managing Editor suggested us to make word counts of our manuscript more than 2500, and reference number more than 25. According to the suggestions, we have also modified our manuscript. “Department of Cancer, Infection and Immunology, Graduate School of Medicine/Faculty of Medicine, Osaka University, Suita City, Osaka, Japan” was added as my affiliation in Line 9-10.
#Comment 1
HP diagnosis should be clarified according to ARS/ERS guideline
#Response 1
We have mentioned radiological findings as compatible with the HP (L76) and pathological findings as probable HP pattern (L86). BAL lymphocytosis was observed (L 77). Etiological antigen was not specified although we have suspected home-related antigens. Hence this patient was diagnosed with FHP with moderate confidence according to ATS/ERS/JRS guideline (L89).
#Comment 2
What is the respiratory exposure.
#Response 2
I have added some information in the Case report section. He worked as an electrician for 10 years in his 20s (L70). Asbestos exposure could not be denied (L 70) (L); however, pleural plaque suggesting asbestos exposure was not noted on Chest CT (L 77). He did not used a feather duvet, and he had no experience raising birds (L74).
#Comments 3
Duration of symptoms and chest abnormality.
#Response 3
The patient was asymptomatic, and chest abnormality was detected firstly.
#Comment 4
Cases of coexisting HP and PAP are not numerous, and the temporality between these two entities is obviously an essential point of discussion.
#Response 4
Two diseases of our patient were simultaneously diagnosed.
We have shown a HP preceding case (L316-L321) and a PAP preceding case in the discussion section (L616-L622).
#Comment 5
Could the authors please specify which biological test they used to quantitatively determine the presence of autoantibodies. Is itthe same technique at the different time points?
#Response 5
We have measured GMAb by ELISA at all time point by the same method (L67-L69).
#Comment 6
Was the screening for autoimmunity associated with interstitial lung disease otherwise negative?
#Response 6
We have added the information.
Serum autoantibodies suggesting connective tissue diseases were not detected (L83-L84).
#Comment 7
Was there a preferential distribution of ground-glass and fibrous areas on the thoracic imaging
#Response 7
As we have shown in the Fig 1, GGO and reticular opacity was predominantly observed in the lower lung field. This information is included in the legend of Figure 1.
#Comment 8
Figure 3: the x-axis date legend is not clear for me.
#Response 8
We have added the x-axis according to your suggestion in figure 3.
#Comment 9
They lack solid arguments to support their hypothesis, due to the retrospective nature of this description. Would it be possible to also measure level of free GM-CSF in BAL?
#Response 9
GM-CSF has been reported to be associated with granuloma formation in vitro and in vivo [13, 14]. GM-CSF concentration in BAL fluids of HP was significantly higher than that in healthy individuals [15].(L 571) However, we could not measure the GMCSF levels in BAL fluid as shown in the limitation (L 825).
#Comment 10
However, this is an attractive hypothesis, and the presentation of this case raises the question of the role of GM-CSF and anti-GMCSF antibodies in the pathophysiology of certain types of interstitial lung disease. The role of anti-GM-CSF antibodies as a regulator of pulmonary GM-CSF levels could indeed be a means of mitigating the onset or aggravation of certain inflammatory ILD.
#Response 10
This is very important point as a reviewer suggested. We have discussed this problem in the discussion (L 737-L758).
#Comment 11
This could open up prospects for systematically measuring GMCSF levels in BAL and anti-GM antibody levels (blood and BAL) in the diagnosis and follow-up of HP. I am less confident about the immediate prospects in terms of treatment (“Deciding on the
appropriate treatment for HP with APAP.”)
#Response 11
As a reviewer suggested, the important for GMAb for treatment decision might not be clarified; however, we suppose better treatment realized by better understanding of pathophysiology. In similar patients as ours whose GMAb was reduced by prednisolone, prednisolone could be used for relapse of the disease without re-evaluation by bronchoscopy, which needs to be proved by additional similar cases are needed.
Reviewer 2 Report
Comments and Suggestions for Authors
Major Comment:
1)
In the case study titled “Serial Measurement of Anti-GM-CSF Autoantibodies in Hypersensitivity Pneumonitis with Autoimmune Pulmonary Alveolar Proteinosis’, Arai and colleagues reported clinical, histological, radiological and serological finding in a patient diagnosed of Hypersensitivity Pneumonitis (HP) with Autoimmune Pulmonary Alveolar Proteinosis (aPAP). They evaluated, through serial measurement, the serum levels of GM-CSF autoantibodies (GMAb), drawing a positive association between GMAb levels and hypersensitivity pneumonitis disease activity. Importantly, they hypothesize that the induction of GMAb may have been an immunological response mechanism to counteract the macrophage-driven HP. The strength of this hypothesis is weakened due to absence of GMAb data during 2 of 3 HP relapses. Although the authors acknowledge this limitation, given the importance of the missing data to the hypothesis proposed, could the authors provide more context and comment on whether the GMAb testing was attempted at those periods during the case progression? This is important for transparency and research integrity. However, the overall trends of the data presented, including low levels of GMAb during HP remission are in agreement with the hypothesis and, to some extent, the authors’ conclusion (further comments on conclusion below).
2)
Despite the limited extrapolation of case studies, the serial measurement of serological parameters over a period of nearly 6 years is very useful for a clinical audience and provides valuable information in relation to the clinical manifestation of the disease. The manuscript describes a potentially new sequence in the coexistence of HP with autoimmune PAP. While this is an interesting hypothesis with important implication for the management of HP and aPAP in specific patients, it will be important and of interest to the readership if the authors were to comment more on alternative hypotheses, present in the literature, for the co-existence of HP and PAP. This is especially important given that their report of GM-CSF reduction following steroid treatment, and their hypothesis that HP in a sense, induced aPAP-associated GM-CSF autoantibody are uncommon. The data presented and the discussions do not exclude the possibility that in this particular patient, a reverse sequence is true. Simply put, could aPAP have appeared before HP in this case as has been previously reported? And if so, this should be clearly stated in the text.
3)
The authors made certain incorrect deductions from some of their references. This suggests a level of bias in interpreting existing literature in the context of their current manuscript. In referring to the publication by Silva et al. (2021), the statement: “In this case, the onset of HP was suppressed by GMAb when APAP was prominent…” (line 181) is incorrect. Silva et al, did not make this claim and nothing in their Letter to the Editor suggest that GMAb suppressed HP. Similarly, the assertion in line 151 that Verma et al. (2010), “hypothesized that their two patients with HP and PAP showed negative serum GMAb because GMAb was neutralized by GM- CSF…” is incorrect. Verma et al. instead distinguished between autoimmune PAP (with GMAb) and secondary PAP due to inhalation exposure (clinically manifesting without GMAb), and suggested that the absence of GMAb in those patients may be due to secondary PAP. The authors must take care to properly represent the work of others and refrain from drawing incorrect conclusions that suit their proposed hypothesis.
A minor comment in relation to line 152 is for the reworded of the sentence to align with the general understanding that antibodies neutralize antigens (or in this case a cytokine) and not a cytokine neutralizing the antibody. However, this comment is overshadowed by the major comment above which faults the authors' interpretation of that section of the paper by Verma et al., (2010).
4)
In line 86, the authors report KL-6 value of 1377 U/ml whereas the data point on the graph which it describes is about 10-fold higher. It is likely that a digit was missed by typographical error in line 86. This changes the result significantly and needs to be corrected.
Minor comments:
1)
It is clear that the authors used PAP to refer to the general condition of pulmonary alveolar proteinosis and APAP to refer to PAP in which GM-CSF autoantibody was present. However, for consistence with other related works in the literature, the authors could use ‘autoimmune PAP’ or ‘aPAP’ instead of ‘APAP’ throughout the manuscript as these abbreviations are more commonly used in the literature. This will also avoid confusion with Acetaminophen which is sometimes abbreviated as APAP due to its chemical name, N acetyl- para-aminophenol.
2)
Line 37 (abstract) and line 198 (conclusion): I will recommend the authors replace ‘sufficient’ with ‘better’ or other appropriate adjectives. Given the complexity of HP and autoimmune PAP, serial GM-CSF autoantibody measurement may not provide sufficient understanding as the manuscript claim but could improve our understanding of the disease pathophysiology and guide appropriate treatment.
3)
To help reader comprehension, the authors should cite appropriate figures for the results described in the text including but not limited to
Line 59: for periodic acid-Schiff test
Lines 61, 87, 93: for GMAb and KL-6 test
4)
Line 175: Authors should provide reference for the role of GM-CSF in granuloma formation
5)
Line 132, 177: Reference in the text to previous publications by multiple authors should include ‘‘et al.” and not just the last name of the first author, in recognition of the contribution of co-authors.
6)
The authors hypothesized that GMAb may suppress granuloma formation in HP. Although this is logical, it is not directly supported by data presented in the manuscript. The authors should therefore back this statement up with citation and discussion of the relevant literature.
7)
To justify the choice of KL-6 for disease evaluation in this study, authors should provide a brief introduction to KL-6 as a biomarker in interstitial lung disease.
Author Response
Response to Reviewer 2
Thank you for your important comments. I have modified according to your comments, and I would like to explain how we have modified point by point.
In addition, a managing Editor suggested us to make word counts of our manuscript more than 2500, and reference number more than 25. According to the suggestions, we have also modified our manuscript. “Department of Cancer, Infection and Immunology, Graduate School of Medicine/Faculty of Medicine, Osaka University, Suita City, Osaka, Japan” was added as my affiliation in Line 9-10.
#Comment 1
Given the importance of the missing data to the hypothesis proposed, could the authors provide more context and comment on whether the GMAb testing was attempted at those periods during the case progression?
#Response 1
This is important point. I am sorry, but we did not know the association between GMAb and clinical course until the measurement of GMAb in all serum samples. As a result., we have missed the serum samples at the important point to explain our hypothesis.
Hence, we have added this problem in the limitation section as below (L791-L793): “In this study, we measured serum levels of GMAb retrospectively except at diagnosis, and unfortunately, serum samples were missed at important time points.”
#Comment 2
Simply put, could aPAP have appeared before HP in this case as has been previously reported? And if so, this should be clearly stated in the text.
#Response 2
Chest abnormal shadow was firstly detected during the health check. He had a no symptoms. Hence, aPAP was not confirmed before the first detection of chest abnormal shadows during the health check.
And I have modified the text as follows (L71-72); A 67-year-old male current smoker presented to our hospital 6 months after the first detection of abnormal chest shadows during a health check.
#Comment 3
Correct description from the previous studies, by Silva and Verma, is needed.
#Response 3
We have modified description and discussion about the paper of Silva in L615-L624.
About the paper of Verma, we have described and discussed in L336-L338.
#Comment 4
Minor comment in relation to line 152 is for the reworded of the sentence to align with the general understanding that antibodies neutralize antigens (or in this case a cytokine) and not a cytokine neutralizing the antibody.
#Response 4
We have modified the part about Verma’s paper, and the sentence of L152 of our original manuscript was deleted.
#Comment 5
In line 86, the authors report KL-6 value of 1377 U/ml where as the data point on the graph which it describes is about 10-foldhigher.
#Response 5
Thank you for pointing the important. We have corrected “1377” to “13770” as you suggested. This was caused by my type error.
#Comment 6
aPAP should be used in the place of APAP.
#Response 6
I have changed APAP to aPAP.
#Comment 7
Line 37 (abstract) and line 198 (conclusion): I will recommend the authors replace ‘sufficient’ with ‘better’ or other appropriate adjectives. Given the complexity of HP and autoimmune PAP, serial GM-CSF autoantibody measurement may not provide
sufficient understanding as the manuscript claim but could improve our understanding of the disease pathophysiology and guide appropriate treatment.
#Response 7
Thank you for your suggestion.
I have changed “sufficient” to “better” in the abstract and conclusion.
#Comment 8
To help reader comprehension, the authors should cite appropriate figures for the results described in the text including but not limited to
Line 59: for periodic acid-Schiff test
Lines 61, 87, 93: for GMAb and KL-6 test
#Response 8
We have added the figures showing PAS positive material in surgical lung biopsy specimens (Fig. 5) and BAL cytology (Fig.2).
We have added the “A” to “H” in the figure of clinical course to show the time point of KL-6 and GMAb in the text (Fig.3).
#Comment 9
Line 175: Authors should provide reference for the role of GM-CSF in granuloma formation
#Response 9
We have added the ref 13, 14 showing GMCSF is associated with granuloma formation (L331).
#Comment 10
Line 132, 177: Reference in the text to previous publications by multiple authors should include ‘‘et al.” and not just the last name of the first author, in recognition of the contribution of co-authors.
#Response 10
We have added the “et al.”
#Comment 11
The authors hypothesized that GMAb may suppress granuloma formation in HP. Although this is logical, it is not directly supported by data presented in the manuscript. The authors should therefore back this statement up with citation and discussion of the relevant literature.
#Response 11
To show the association between GMAb and granuloma formation, we have added the paragraph. (L752-L773). We have reported activity of sarcoidosis and tuberculous lymphadenitis might be associated with GMAb levels.
#Comment 12
To justify the choice of KL-6 for disease evaluation in this study, authors should provide a brief introduction to KL-6 as a biomarker in interstitial lung disease.
#Response 12
We have added the paragraph showing the importance of KL-6 as a biomarker of ILDs and PAP. (L757-L771).
Round 2
Reviewer 2 Report
Comments and Suggestions for Authors
In the case study titled “Serial Measurement of Anti-GM-CSF Autoantibodies in Hypersensitivity Pneumonitis with Autoimmune Pulmonary Alveolar Proteinosis’, I am satisfied that the authors, Arai and colleagues, addressed all the comments raised at the initial review. The manuscript has been sufficiently improved to warrant publication.
There are a few minor typographical errors which I wish to bring to the attention of the editor and authors for correction. They include but are not limited to the list below.
- Line 50: missing word
- Line 60: grammar
- Line 222: grammar
- Line 236/237: grammar
Except for the a few typographical and grammatical errors, the quality of English Language in the manuscript is good.
Author Response
Response to Reviewer 2
Thank you for your important comments. I have modified according to your comments, and I would like to explain how we have modified point by point. Modified points were highlighted by green color.
In addition, we have found mistakes in reference number.
Reference number 8 in line 68 was changed to 9.
Reference number 20 in line 272 to 26.
#Comment 1
There are a few minor typographical errors which I wish to bring to the attention of the editor and authors for correction. They include but are not limited to the list below.
Line 50: missing word
Line 60: grammar
Line 222: grammar
Line 236/237: grammar
#Response 1
Thank you for pointing some errors. I have corrected the mistakes as follows.
Line 50:
“several patients with HP complicated by have been reported [6,7]” was changed to
“several patients with HP complicated by aPAP have been reported [6,7]”
Line 60: grammar
“He did not used a feather duvet, and he had no experience raising birds” was changed to
“He did not use a feather duvet, and he had no experience raising birds”.
Line 222: grammar
“though GM-CSF inhalation is now be favored [20]” was changed to
“though GM-CSF inhalation is now favored [20]”.
Line 236/237: grammar
“Hence the patient was diagnosed with FHP and aPAP after the multidisciplinary discussion. The pathophysiology of HP development of after the control of aPAP has not been clarified.”
Changed to
“Hence, the patient was diagnosed with FHP and aPAP after the multidisciplinary discussion. The pathophysiology of HP development after the control of aPAP has not been clarified.”
